



# An Arctic watershed observatory at Lake Peters,
# Alaska: weather-glacier-river-lake system data for
# 2015-2018
**Authors**
Ellie Broadman[1], Lorna L. Thurston[2], Nicholas P. McKay[1], Darrell S. Kaufman[1], Erik Schiefer[1],
David Fortin[3], Jason Geck[4], Michael G. Loso[5], Matt Nolan[6], Stéphanie H. Arcusa[1], Christopher
W. Benson[1], Rebecca A. Ellerbroek[1], Michael P. Erb[1], Cody C. Routson[1], Charlotte Wiman[1], A.
Jade Wong[1]
1. School of Earth and Sustainability, Northern Arizona University, Flagstaff, Arizona, 86011,
USA
2. Stillwater Sciences, 2855 Telegraph Ave, Ste 400, Berkeley, California, 94705, USA
3. Department of Geography and Planning, University of Saskatchewan, Saskatoon, SK S7N
5C8, Canada
4. Environmental Science Department, Alaska Pacific University, Anchorage, Alaska, 99508
5. Inventory and Monitoring Program, Wrangell-St. Elias National Park and Preserve, Copper
Center, Alaska, USA
6. Fairbanks Fodar, Fairbanks, Alaska, USA





# Abstract

Datasets from a four-year monitoring effort at Lake Peters, a glacier-fed lake in Arctic Alaska,
are described and presented with accompanying methods, biases, and corrections. Three
meteorological stations documented air temperature, relative humidity, and rainfall at different
elevations in the Lake Peters watershed. Data from ablation stake stations on Chamberlin
glacier were used to quantify glacial melt, and measurements from two hydrological stations
were used to reconstruct continuous discharge for the two primary inflows to Lake Peters,
Carnivore and Chamberlin Creeks. The lake's thermal structure was monitored using a network
of temperature sensors on moorings, the lake's water level was recorded using pressure
sensors, and sedimentary inputs to the lake were documented by sediment traps. We
demonstrate the utility of these datasets by examining a flood event in July 2015, though other
uses include studying intra- and inter-annual trends in this weather-glacier-river-lake system,
contextualizing interpretations of lakes sediment cores, and providing background for modeling
studies. All DOI-referenced datasets described in this manuscript are archived at the National
Science Foundation Arctic Data Center at the following overview webpage for the project:
https://arcticdata.io/catalog/view/urn:uuid:517b8679-20db-4c89-a29c-6410cbd08afe (Kaufman
et al., 2019e).

# 1. Introduction

Arctic glacier-fed lakes are complex and dynamic systems that are influenced by diverse
physical and biological processes. Long-term instrumental datasets that document how weather
and climate impact glaciers, basin hydrology, and sediment transport through rivers and lakes
are rare, especially in the Arctic. Such datasets are critical for studying how Arctic glacier-river-
lake catchments operate as a system, for understanding the processes that control sediment



accumulation in lakes, and for contextualizing modern climatic and environmental change
relative to past centuries.

Here we present the results of a four-year instrumentation campaign in the watershed of Lake
Peters, a large glacial lake located in the northeastern Brooks Range within Alaska's Arctic
National Wildlife Refuge. Lake Peters is a particularly valuable site for such an observatory due
to the variety of components influencing processes within the lake, including both a relatively
small, glacially-dominated inflow (Chamberlin Creek) as well as a larger primary inlet with less
glacier coverage (Carnivore Creek). A research station at Lake Peters was first established in
the late 1950's by the Department of the Navy as a substation of the Barrow Naval Arctic
Research Laboratory, and is maintained as an administrative facility by the U.S. Fish and
Wildlife Service. In 2015, an array of instruments were installed and observational data
collection commenced for a variety of components of the weather-glacier-river-lake system (Fig.
1). All instrument installations were non-permanent in recognition of the Wilderness status of the
study area.

By collecting meteorological, glaciological, fluvial, laucustrine, spatial, and geochemical data
within the Lake Peters catchment, interconnections in the local hydrologic system can be
quantified and explored in detail. Additionally, this multi-year monitoring study provides useful
context for the interpretation of past lake sediments from Lake Peters (Benson, 2018; Benson et
al., in review). In the interest of making these unique and valuable data readily accessible to
potential users, the objective of this manuscript is to document and describe the datasets, as
well as data collection methods, biases, and corrections.



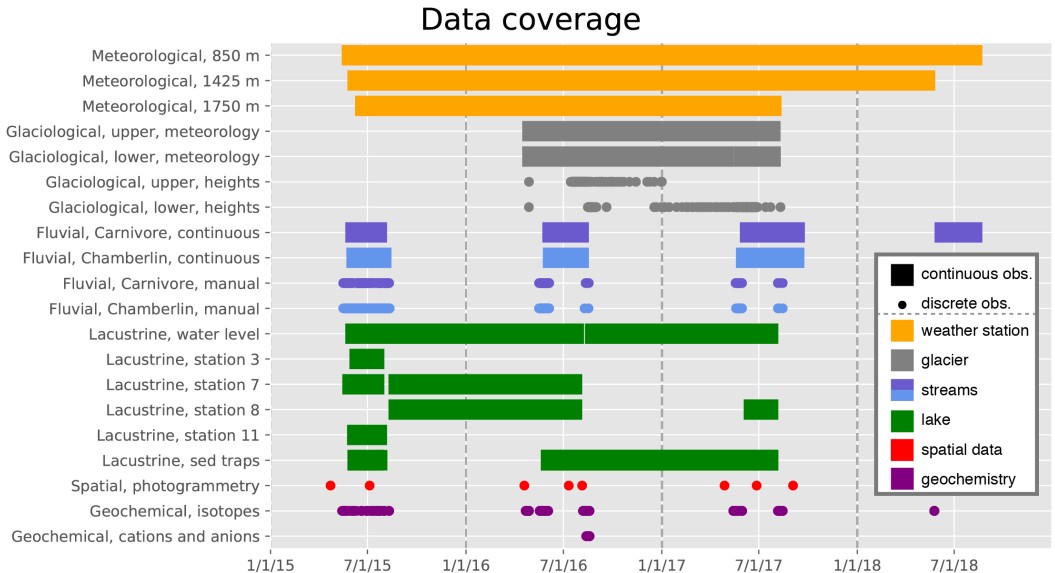


Figure 1. Temporal coverage of data collected in the Lake Peters catchment. Data are either

continuous hourly or bi-hourly observations (bars) or discrete observations in time (dots). Data

are indicated as available if any variable is present for the given time. Sediment traps are not

hourly, but were collected over specific windows of time. January 1st of each year is marked

with a dotted line. Data collected from different parts of the Lake Peters catchment are shown

with different colors, in roughly the same order as they are discussed in Section 3. See Section

3 for a more detailed description of the data. Some data (Lake Peters TROLL casts and

bathymetry) are not represented in this figure.

# 2. Site description

Lake Peters (69.32°N, 145.05°W) lies at 853 meters above sea level (m a.s.l.) on the north side
of the Brooks Range, an east-west trending mountain range in Arctic Alaska. The Brooks Range
extends 1000 km, separating the Arctic coastal plain to the north from the Yukon Basin to the



south. The third tallest peak in the range, Mount Chamberlin (2712 m a.s.l.; Nolan and
Deslauriers, 2016) lies 3 km east of Lake Peters.

One of the largest glacier-fed lakes in Arctic Alaska, Lake Peters has a maximum water depth of
52 m and an area of ~6.4 km$^2$. The lake's catchment (171 km²) has 8% glacier coverage based
on our aerial photography in 2016, and receives a majority of stream flow from Carnivore Creek,
which has a 128 km$^2$ catchment with 10% glacier coverage (Fig. 2). The Carnivore catchment
contains 6 glaciers exceeding 0.5 km² in area and numerous smaller glaciers, with a median
elevation for all glaciers at 1910 m a.s.l. The largest glacier (3.5 km²; 1410 - 2430 m a.s.l.) is
situated at the head of Carnivore Creek. Since the Little Ice Age, the largest few catchment
glaciers have reduced in length by over 30%. Lake Peters' secondary inflow, Chamberlin Creek,
has a 8 km$^2$ catchment that is 21% covered by Chamberlin Glacier (1.7 km²; 1600 - 2710 m
a.s.l.) (Fig. 2). The lake drains to the north into Lake Schrader, whose outflow, the Kekiktuk
River, discharges into the Sadlerochit River and ultimately the Arctic Ocean. Lake Peters and
Lake Schrader, or "Neruokpuk Lakes", are ice covered from early October through mid-June or
July, and often thermally stratify (Hobbie, 1961). The basin is north of the modern tree line in a
region underlain by continuous permafrost, with a thin active layer where soils are rocky,
excessively drained, strongly acidic, and have very thin surface organics. Bedrock within the
basin is Devonian - Jurassic sedimentary to meta-sedimentary rocks primarily of the Neruopuk
Formation (Reed, 1968).

Climate at Lake Peters is semi-continental, with temperatures more extreme than locations on
the coastal plain. A short observational time series of weather from Lake Peters was initiated
during the International Geophysical Year (Hobbie, 1962), and was summarized by March
(2009). For the year between May 1, 2015 and April 30, 2016, weather stations installed in Lake
Peters basin recorded an hourly averaged temperature of -9.8°C and a liquid precipitation total





of 166 mm. During this time period, average temperature was -29°C in December and 8.5°C in
July. Strong temperature inversions develop during the winter, with differences as large as 15°C
between 850 and 1425 m a.s.l.

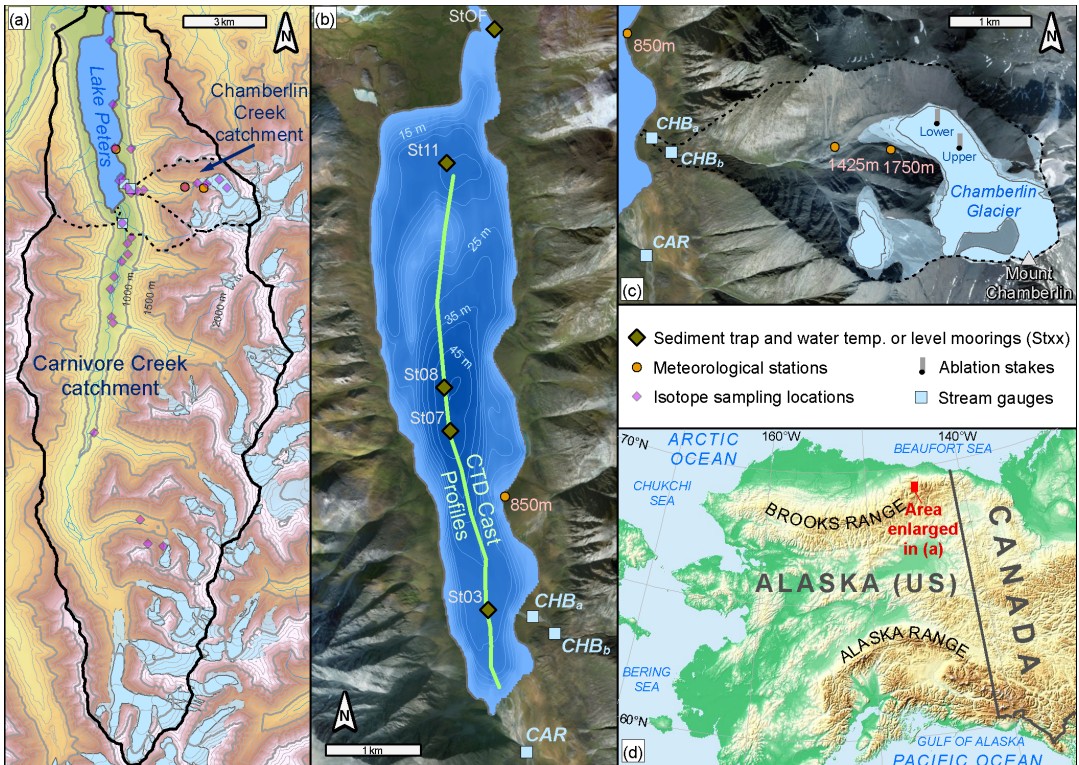


Figure 2. (a) Lake Peters catchment (solid black boundary) and Carnivore and Chamberlin
Creek sub-catchments (dashed boundaries) with terrestrial data sampling locations. (b) Lake
Peters bathymetry and sampling locations. (c) Chamberlin Creek sub-catchment and
Chamberlin Glacier. (d) Brooks Range setting in Alaska and study area location.




# 116    3. Methods

## 117    3.1 Meteorological data

Three weather stations operated in the catchment from May 2015 through August 2018 along
an elevational gradient (Fig. 3) (Kaufman et al., 2019c). The lowest elevation station was
mounted on the roof of the cabin at the research station at 850 m a.s.l. near Lake Peters.
Additional stations were installed on tripods at 1425 m a.s.l. inside the Little Ice Age moraine of
Chamberlin Glacier, and the highest elevation station was at 1750 m a.s.l. at the crest of the left
lateral moraine of Chamberlin Glacier (Fig. 3).

All three stations were equipped with air temperature and relative humidity sensors (Hobo Pro
v2 Temp/RH sensors) and ground temperature sensors at 2 and 30 cm depth (Hobo Pro V2 2X
w/ 6ft extension). The 850 and 1425 m a.s.l. stations were equipped with backup air
temperature and relative humidity sensors (Hobo Pro v2 Temp/RH sensors). All temperature
and relative humidity sensors were housed in Hobo Solar Radiation Shields. The 850 m a.s.l.
station was additionally equipped with a barometer (Hobo Barometric Pressure Sensor S-BPB-
CM50), an anemometer (Young Wind Monitor - 5103), and a pyranometer (Solar Radiation
Sensor - S-LIB-M003). At all three stations, tipping bucket rain gauges (Hobo Rain gauge
0.2mm w/pendant - RG3-M) were installed on the ground away from the stations and equipped
with Alter-type windshields to reduce wind-related undercatch. Correlations between daily
precipitation recorded at 850 m a.s.l. with daily precipitation recorded at 1425 and 1750 m a.s.l.
indicate that trends are regionally coherent within the watershed with minimal (but expected)
differences (Fig. 4).



All three stations were generally continuously operational, though instrument failure caused
gaps in the collection of some data from September 21, 2015 to April 19, 2016 at the 850 m
a.s.l. station. Shorter instrument failures are documented for each sensor in the data files.

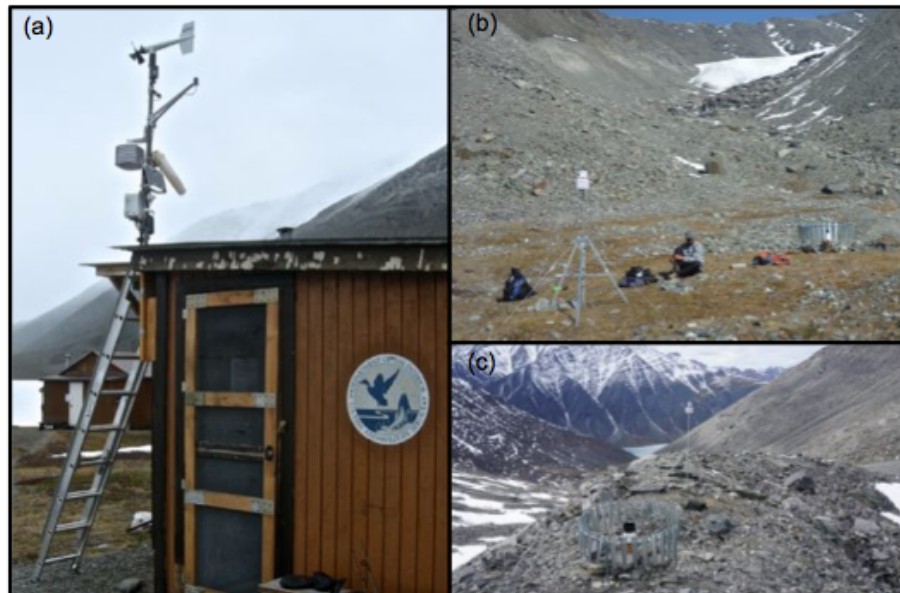


Figure 3. Weather stations at (a) 850, (b) 1425, and (c) 1750 m a.s.l. in the Lake Peters
watershed.


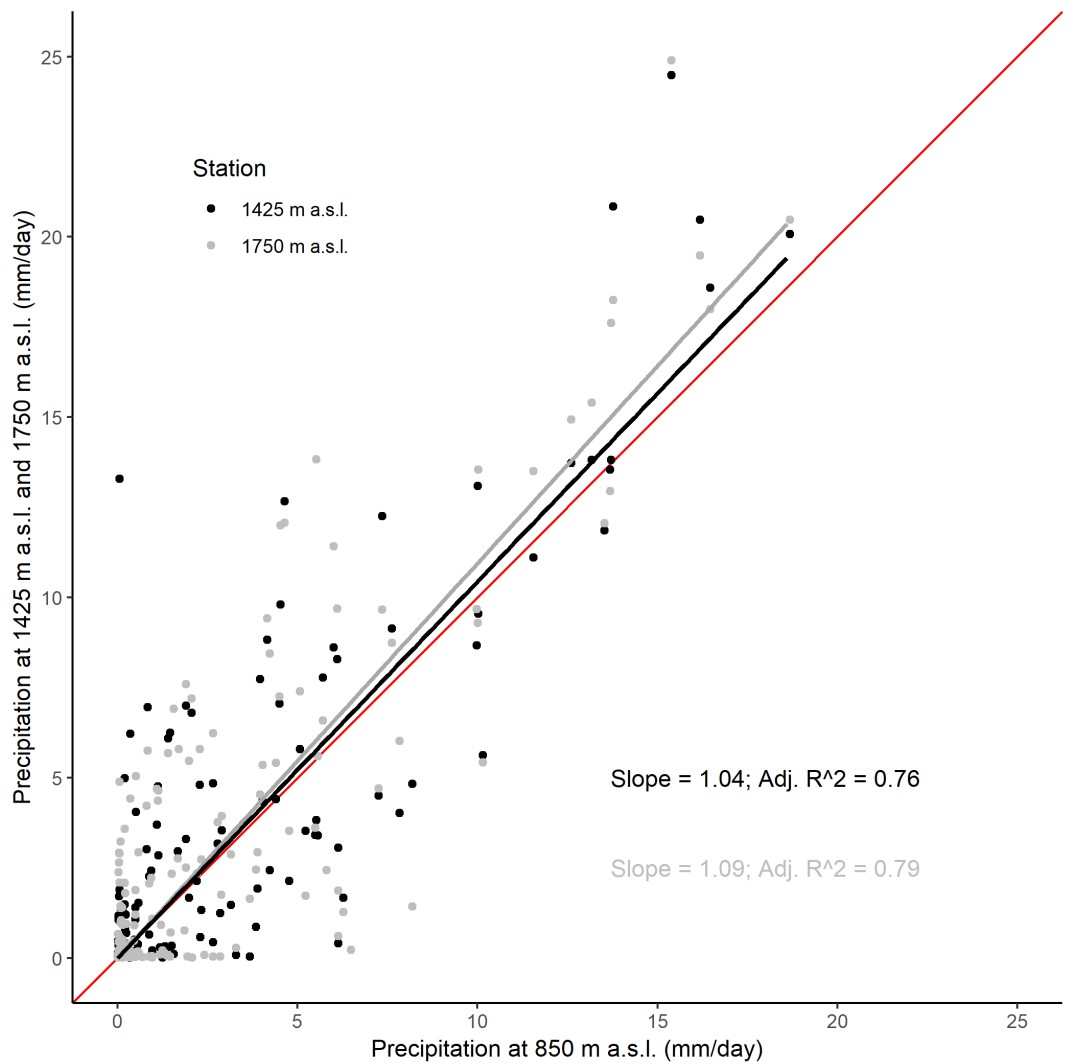


Figure 4. Correlation of daily precipitation at 850 m a.s.l. with daily precipitation at 1425 and
1750 m a.s.l..

## 151   3.2 Glaciological data

The elevation of the surface of Chamberlain Glacier and its seasonal snow cover was observed
using ablation stakes and interval photography (Geck et al., 2019). Graduated stakes were



installed on April 27, 2016 at two sites along the lower part of the glacier: lower (69.29312°N,
144.93814°W; 1772 m a.s.l.) and upper (69.29031°N, 144.93134°W; 1860 m a.s.l.). (Fig. 5a).
Photographs captured exposed stakes height using Wingscapes Time-lapse Cameras (SKU:
WCT-00122) set to automatically record every 6 hours. Ablation stake heights were observed
on photographs (April 27, 2016 - August, 11, 2017) at a precision of 1 cm. Cameras were
mounted on tripods constructed of electrical conduit (Fig. 5b). Air temperature sensors (Hobo
Pro v2 Temp/RH sensors and/or Hobo Pendant) housed in Hobo Solar Radiation Shields
recorded mean hourly temperatures 2 m above the snow surface at each site (Fig. 5b)
(Kaufman et al., 2019f).

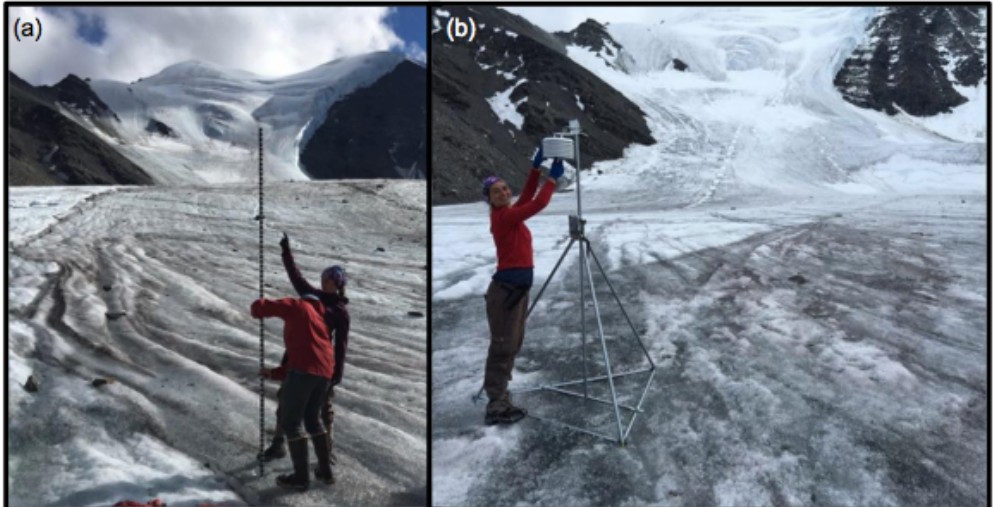


Figure 5. Examples of (a) ablation stakes (lower site) and (b) weather stations (upper site)
installed on Chamberlin Glacier.

## 167  3.3 Fluvial data

Two hydrological stations were established in May 2015: one at Carnivore Creek (Schiefer et
al., 2019b) and one at Chamberlin Creek (Schiefer et al., 2019c). At each station, an In-Situ





TROLL 9500 (TROLL) was deployed during the open-channel season to measure pressure,
temperature, conductivity, and turbidity, and a Hobo Onset U20 Water Level logger was used
for backup measurements of pressure and temperature. The instruments were attached to a
cage constructed from aluminum tubing and chicken wire. A stage gage was installed at each of
the hydrological stations, and water level and reach conditions were captured in hourly
photographs by field cameras. Cages were installed in thalwegs of stable reaches and loaded
with boulders to prevent movement (Fig. 6a). In August 2015 the instruments were detached
and data were downloaded. In Carnivore Creek, the same procedure was completed for the
2016, 2017, and 2018 field seasons (May-August), though data were not successfully retrieved
from the TROLL in 2018. In Chamberlin Creek, the cage shifted several times during the 2015
field season, and the instruments were consequently secured to channel-side bedrock for the
2016 field season in an upstream pool near the alluvial fan apex (Fig. 6b). The instruments were
ultimately carried downstream in the largest flood event of 2016, which peaked on July 16, and
were subsequently secured to a large boulder for the 2017 field season, near the original 2015
gauging station. No instruments were successfully deployed in Chamberlin Creek in 2018. In
addition to these continuous datasets, discrete discharge (Q) and suspended sediment
concentration measurements were taken using a hand-held Hach FH950 flow meter and a US
DH-48 suspended sediment sampler, respectively.

Water pressure was corrected using barometric pressure measurements from the 850 m a.s.l.
meteorological station and then converted to stage. Discrete field sampling of discharge was
most frequent and spanned the greatest range in 2015. Discharge–stage regressions were used
to construct continuous half-hourly discharge for the majority of the 2015 open-channel season
in both Carnivore and Chamberlin Creeks (Fig. 7a-b). In Carnivore Creek, photographs of water
level were imported to image viewing and analysis software to estimate 2.6 days of peak stage
during a flood on August 3, 2015 (Fig. 7c) and its error margin. The regression used nine



photographs selected to capture maximal water-level variability, and stage was established
using a point-to-point method to capture temporal variability. In Chamberlin Creek, linear
relations between stage from the Hobo and TROLL instruments permitted compilation of a
seamless discharge record for 2015 (Fig. 7d).

For the 2016 open-channel season, alternative methods were applied due to the lack of velocity
data required to compute discharge, assuming consistent discharge-stage relations between
seasons (Thurston, 2017). To formulate discharge for Carnivore Creek, stage was first
regressed against cross-sectional area for both open-channel seasons separately (Fig. 7e).
2016 stage was adjusted upward to be consistent with 2015 stage, accounting for the shift in
instrument positioning between years. The 2015 stage-discharge relation could then be used to
establish continuous 2016 season discharge from stage measured in 2016 (Fig. 7a). This
method could not be applied to Chamberlin Creek because the TROLL and Hobo instruments
were secured in different reaches between 2015 and 2016. Instead, 2015 discharge was
regressed against cross-sectional area (Fig. 7f), and this relation was then used to construct
discrete 'sampled' 2016 discharges from 2016 cross-sectional areas. Discharge-stage relations
were used to formulate continuous discharge until a flood event beginning on June 20, 2016
(Fig. 7g), at which point the instrumentation came loose and sensors failed. Several methods
were used to construct stage and discharge for Chamberlin Creek following this flood (Fig. 7h-j).
These methods included: photographs of river level from time-lapse cameras (46 days of
discharge values); a regression of Chamberlin and Carnivore Creek discharges using all
available data (3.6 days); a three-point stage-discharge rating curve, sampled downstream of
the rating curve applied prior to the flood (8.8 days). When none of the aforementioned
alternatives were possible, gaps were interpolated, amounting to a total of five gaps that range
from 2 to 19 hours in length. The majority of the 2016 stage record (June 20 - August 5, 2016)
was estimated for Chamberlin Creek from photographs of water level (Fig. 7h; Thurston, 2017).



Water levels were measured using two reference points on a total of 13 photographs, then
regressed separately against discharge. The first reference point was suitable for most
discharges, but was not accurate after July 16, 2016 when water levels dropped low enough to
expose a small bar. The second reference point was suitable for low discharges, but overtopped
during high flows. Error was conservatively 10-20 image pixels, where 10 pixels equate to an
approximately 0.14 $m^3$ $s^{-1}$ error margin in either direction from the average Q of 0.72 $m^3$ $s^{-1}$. For
the peak of this flood event on July 8, 2016 at 2:00, error is estimated to be 100 pixels in either
direction because water level and control points were obscured (equating to a maximal error
range of 7.31 $m^3$ $s^{-1}$ to 49 $m^3$ $s^{-1}$ for the peak Q of 21 $m^3$ $s^{-1}$). The Chamberlin station was
ultimately re-established in a new location, and a new rating curve was used to reconstruct Q
from August 8, 2016 at 18:00 until August 17, 2016 at 13:00 (Fig. 7i). For Carnivore Creek data,
the photographic method was favored over regression whenever photographs were available to
keep the sub-catchment Q records independent. Considerable scatter between Carnivore and
Chamberlin Creek Q records is attributed to hydrological differences between the sub-
catchments (Fig. 7j). Summarized Q data alongside meteorological data at the 850 m a.s.l.
station are shown in Fig. 8.

Few manual measurements of discharge or suspended sediment concentration were made in
2017 and 2018. Stage-discharge rating curves developed from 2015 and 2016 data were used
to convert continuous stage records to discharge for Carnivore Creek for 2017 and 2018, and
Chamberlin Creek for 2017. Only minor adjustments were made, primarily to account for
changes in instrumentation position relative to the channel bed. Instrumentation failure
prevented development of a 2018 discharge record for Chamberlin Creek.



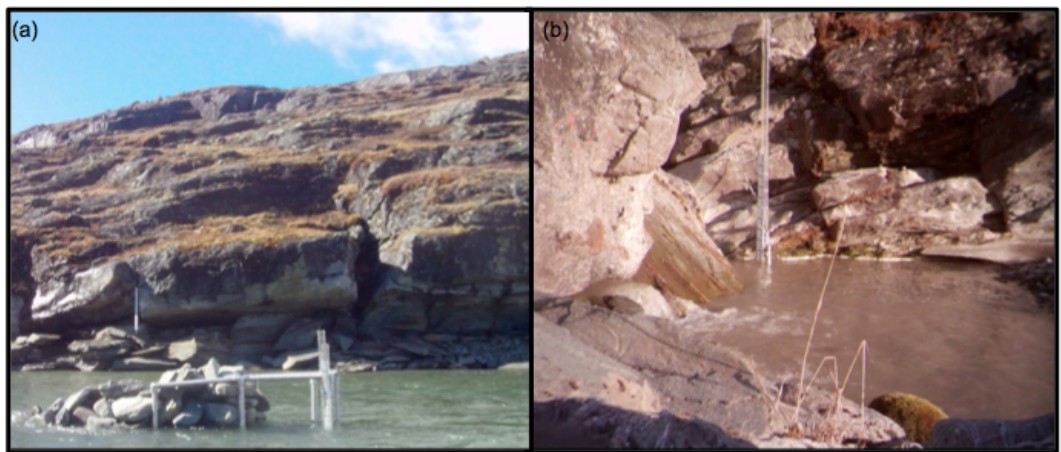


Figure 6. Stream stage gauges for (a) Carnivore and (b) Chamberlin Creeks in June 2016.
Instrumentation mounted in PVC tubing is obscured by stage gages.

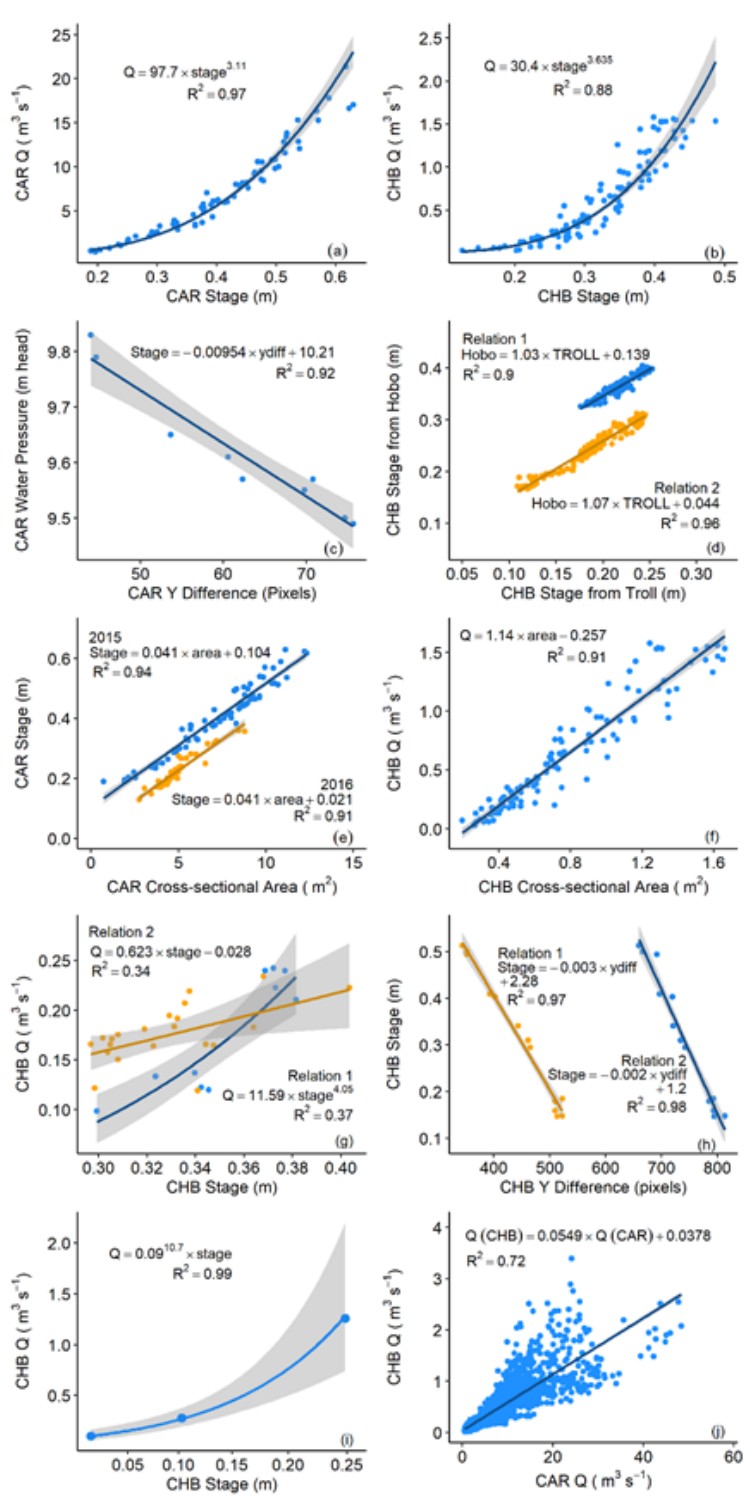




Figure 7. Regression relations and associated confidence bands used to create continuous
discharge (Q) time-series throughout the 2015-2018 open-channel seasons in Carnivore (CAR)
and Chamberlin (CHB) Creeks. (a) Q – stage relation for CAR (2015); (b) Q – stage relation for
CHB (2015); (c) water-pressure (from TROLL) – y difference (y-diff; shown on photographs)
relation for CAR (2015); (d) relations of stage calculated from Hobo U20 water-pressure against
stage calculated from TROLL 9500 water-pressure for CHB (both 2015); (e) stage – cross-
sectional area linear relation for CAR (2015 and 2016); (f) Q – cross-sectional area linear
relation for CHB (2015); (g) Q – stage (from Hobo) power relation for CHB (2016); (h) Stage – y
difference (y-diff; shown on photographs) linear relations for CHB (both 2016); (i) Q – stage
exponential relation for CHB (2016); and (j) Relation of Q in CHB against Q in CAR with a 2-
hour lag (2015 and 2016). Refer to Thurston (2017) for additional information.





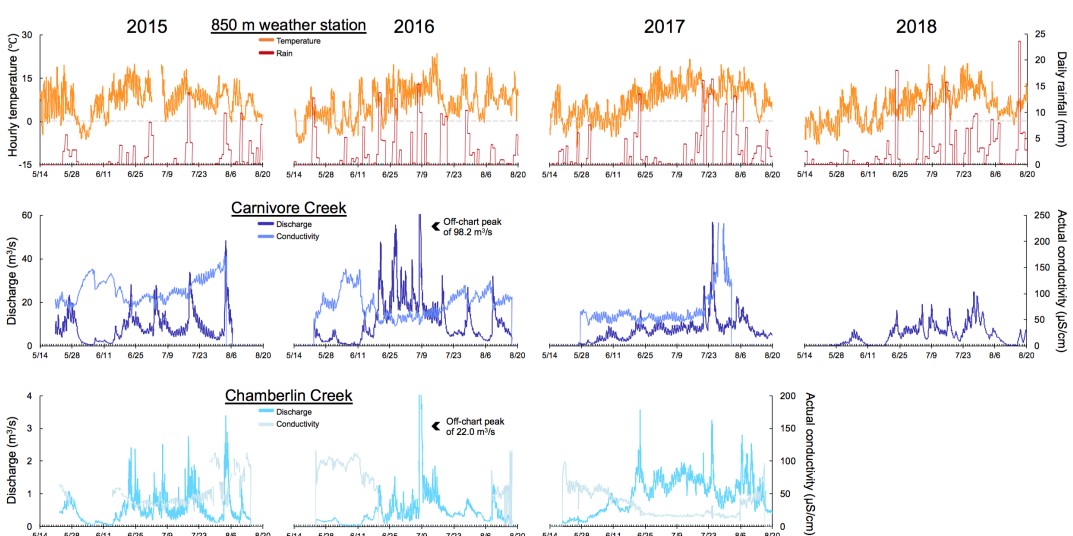


Figure 8. Time series of hourly air temperature and rainfall from the 850 m a.s.l. weather station
at Lake Peters shown alongside conductivity and discharge for Carnivore and Chamberlin
Creeks, 2015-2018.

## 273    3.4 Lacustrine data

### 274    3.4.1 Limnological data

Hourly water level measurements were taken using Hobo U20L Water Level loggers installed on
anchored moorings near the water's surface at two locations in Lake Peters: one in its central
basin and one near its outflow (Fortin et al., 2019a). Pressure data were corrected to water
depth using barometric pressure measurements from the 850 m a.s.l. meteorological station.

Throughout the 2015 field season, a series of TROLL cast transects were taken to provide
temperature, turbidity, and conductivity data at stations along a north-south transect throughout





Lake Peters (Fortin et al., 2019b). Pressure measurements from the TROLL were converted to
water depth, providing vertical profiles for these data. These casts were performed at a varying
number of stations on a total of 33 days over the course of the field season.

In 2015, 2016, and 2017, lake temperature and luminous flux data were collected at various
water depths on anchored moorings throughout the lake (Fig. 2) (Kaufman et al., 2019b). The
moorings were equipped with Hobo pendants and Hobo Water Temp Pro loggers at different
water depths to monitor the thermal structure of the lake. Moorings located in the central basin
of the lake were deployed from May 2015 until August 2016 (Station 7) and August 2015 until
August 2017 (Station 8). A mooring in the distal basin (Station 11) was deployed from May 2015
to August 2015, and a mooring in the proximal basin (Station 3) was deployed from May 2015
until August 2015.

## 3.4.2 Sediment trap data
Throughout 2015-2017, sediment traps were deployed in Lake Peters to measure the rate of
deposition and collect suspended sediments in the lakes (Kaufman et al., 2019d). In 2015, three
pairs of static (i.e., non-automated) traps with different aspect ratios (different collection-tube
heights but same collection-tube diameters) (Fig. 9a) were deployed from May-August at a
central location of Lake Peters to determine the effect of these proportions on sediment
collection efficiency. From May-August 2016 and again from August 2016 through August 2017
one static trap was deployed in Lake Peters. This trap consisted of ordinary 2 L plastic bottles
with the bottoms removed, a 50 ml centrifuge tube secured to its mouth, and inverted to funnel
sediments into the tubes (Fig. 9b). Each trap comprised two or three replicate tubes/bottles
deployed at different depths along an anchored mooring. In addition, an incrementing sediment




trap described by Muzzi and Eadie (2002) was deployed from May 2016 through August 2017,
collecting sediments in 23 bottles rotating over daily to weekly increments (Fig. 9c).
For all sediment samples, dry sediment mass, daily flux, and annual flux were calculated. For
several static trap samples, grain size data (mean, standard deviation, d50, d90, %sand, %silt,
and %clay) were determined using a Coulter LS-320. Samples were pretreated with 30% $H_2O_2$
and heated overnight at 50°C to remove organic material, and then treated with 10% $Na_2CO_3$ for
5 hours to remove biogenic silica. Pretreated samples were deflocculated by adding sodium
hexametaphosphate and shaking for one hour before performing grain size analysis. Organic
matter content for some samples was determined using loss on ignition analysis.

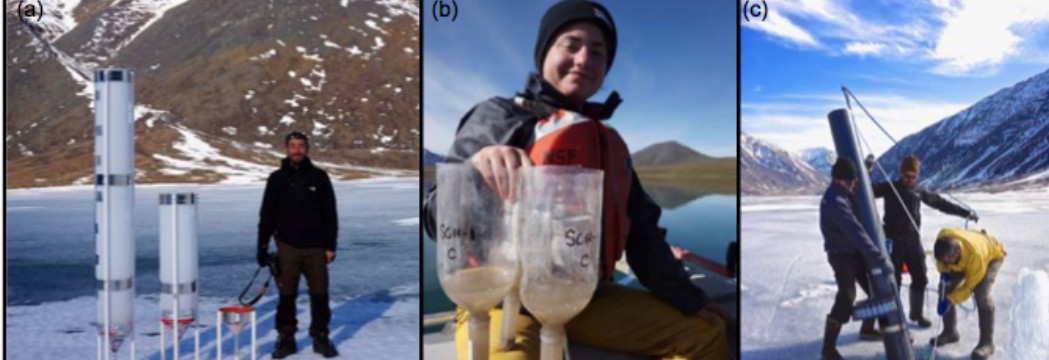


Figure 9. Examples of the types of sediment traps installed in Lake Peters: (a) installation of
static traps ratio traps, (b) static traps upon retrieval, and (c) installation of an incrementing trap.
## 3.5 Spatial data
3.5.1 Bathymetry



Bathymetric data were collected using a Lowrance Elite-5 HDI Combo sonar depth recorder
along several transects throughout Lake Peters in August 2015 (Schiefer et al., 2019a). These
data were used to create a bathymetric map with ArcGIS 9.1.

3.5.2 High resolution photogrammetry
Airborne photogrammetric data were acquired two times in 2015 (April 23 and July 5), three
times in 2016 (April 19, July 11, and August 5), and three times in 2017 (April 28, June 27, and
September 3) (Nolan, 2019). These data were used to create digital elevation models (DEMs)
with postings of 50-80 cm provided in NAD83(2011) NAVD88 Geoid 12B. When conditions
permitted (April 23, 2015; July 5, 2015; April 19, 2016; April 28, 2017), the entire Carnivore
Creek watershed was acquired. On all other dates, only the eastern side of the valley, where the
catchment glaciers are located, was captured. These data were acquired using a Nikon DSLR
attached to a survey grade GNSS on board a manned aircraft, providing photo-center position
to 10 cm or better and thus eliminating the need for ground control. Accuracy within this steep
mountain environment was found to exceed 20 cm horizontally and vertically (Nolan and
Deslauriers, 2016), without use of ground control. Data acquisition methods are described in
detail in Nolan et al. (2015). These DEMs were used to create difference DEMs that can be
used to measure snowpack thickness throughout the entire watershed as well as glacier surface
elevation change. The accuracy of these DEM-derived snow thickness measurements has
previously been found to be comparable to those acquired using hand probes (Nolan et al.,
2015). However, larger errors are expected in rugged mountain topography, as the 50 cm DEMs
cannot resolve many small crenulations, likely causing some spatial biasing.



## 3.6 Geochemical data

A total of 187 water samples were collected from the Lake Peters catchment between 2015 and 2018 and analyzed for isotopes of oxygen ($\delta^{18}$O) and hydrogen ($\delta$D) (‰VSMOW) (Kaufman et al., 2019a). These data, combined with conductivity measurements, were input to a hydrograph mixing model to estimate the relative contributions of various water sources to Carnivore and Chamberlin Creeks (Ellerbroek, 2018). Sampling intervals varied by location but were approximately every 2-6 days when a field team was present. In one 24-hour period in August 2016, each creek was sampled every two hours using a Teledyne ISCO 3700 portable automatic water sampler (ISCO). These ISCO samples from Carnivore and Chamberlin Creeks were also analyzed for major cation and anion geochemistry analysis using a Dionex ICS-3000 ion chromatograph with an analytical precision of ±5% (Kaufman et al., 2019a). Water samples were also collected from three unnamed glaciated tributaries in the Carnivore Creek valley and three unnamed non-glacial streams that intermittently drain into Lake Peters. All samples were collected in 10 mL polyethylene bottles rinsed three times with sample water.

Winter precipitation samples were collected from multiple locations. Twenty-seven snowpack samples were collected with snow density measurements in April 2016 and May 2017 at a variety of depths from 21 snow cores along an elevational gradient. Samples of glacial melt, glacial ice, and snow drifts were collected over 2015 and 2016. All snow and ice samples were collected in airtight plastic bags and transferred into rinsed 10 mL polyethylene bottles after melting. Rain samples were collected sporadically after storm events from spill off from the roof of G. WIlliam Holmes research station and transferred into rinsed 10 mL polyethylene bottles.



Following each field season, water samples were processed using Wavelength-Scanned Cavity
Ringdown Spectroscopy. Analytical precision is approximately 0.3‰ for $\delta^{18}$O and 0.8‰ for δD
(Los Gatos Research, 2018).

# 4. Data example: Tracing an event through the Lake
# Peters glacier-river-lake system
To examine how precipitation works its way through the Lake Peters catchment system, and to
depict a small subset of the available data, multiple datasets are shown for a precipitation event
in the latter half of July 2015 (Figs. 10, 11). During this event, the 850 m a.s.l. meteorological
station recorded a total rainfall of 22.4 mm between July 17-20 (Fig. 10a). In response,
hydrological stations at both Carnivore and Chamberlin Creeks recorded an increase in
discharge to Lake Peters, with Carnivore Creek discharge reaching far higher values due to its
larger catchment (Fig. 2). Discharge then decreased over several weeks, long after the
precipitation event concluded, with diurnal fluctuations illustrating the effect of glacial melt (Fig.
10b). Turbidity increased in both creeks, coincident with the beginning of elevated discharge
(Fig. 10c). The influx of water increased the water level of Lake Peters, which lagged, likely as a
function of the total integration of new water as well as outflow into Lake Schrader (Fig. 10d).
This event had several additional effects on the water of Lake Peters: Turbidity in Lake Peters
increased during the event (Fig. 10e), the inflow of water caused a temporary drop in water
temperatures closer to the lake surface (Fig. 10f), and the amount of luminous flux a few meters
below the water's surface dropped (Fig. 10g). Isotopic values were also recorded from rainfall
and creekwater at several moments during this event (Fig. 10h).



To quantify the spatial characteristics of the sediment pulse entering Lake Peters during this
precipitation event, data from TROLL casts taken throughout the lake on multiple days before,
during, and after the event were examined (Fig. 11). These data indicate that the spatial pattern
of turbidity anomalies throughout Lake Peters is more complex than that captured at a single,
proximal station shown in Fig. 10e.

This multifaceted perspective of a precipitation event represents only a small fraction of the total
data produced by this field effort, but illustrates both the interconnectedness of processes in the
Lake Peters catchment and the potential for these datasets to further our understanding of
Arctic weather-glacier-river-lake systems dynamics.





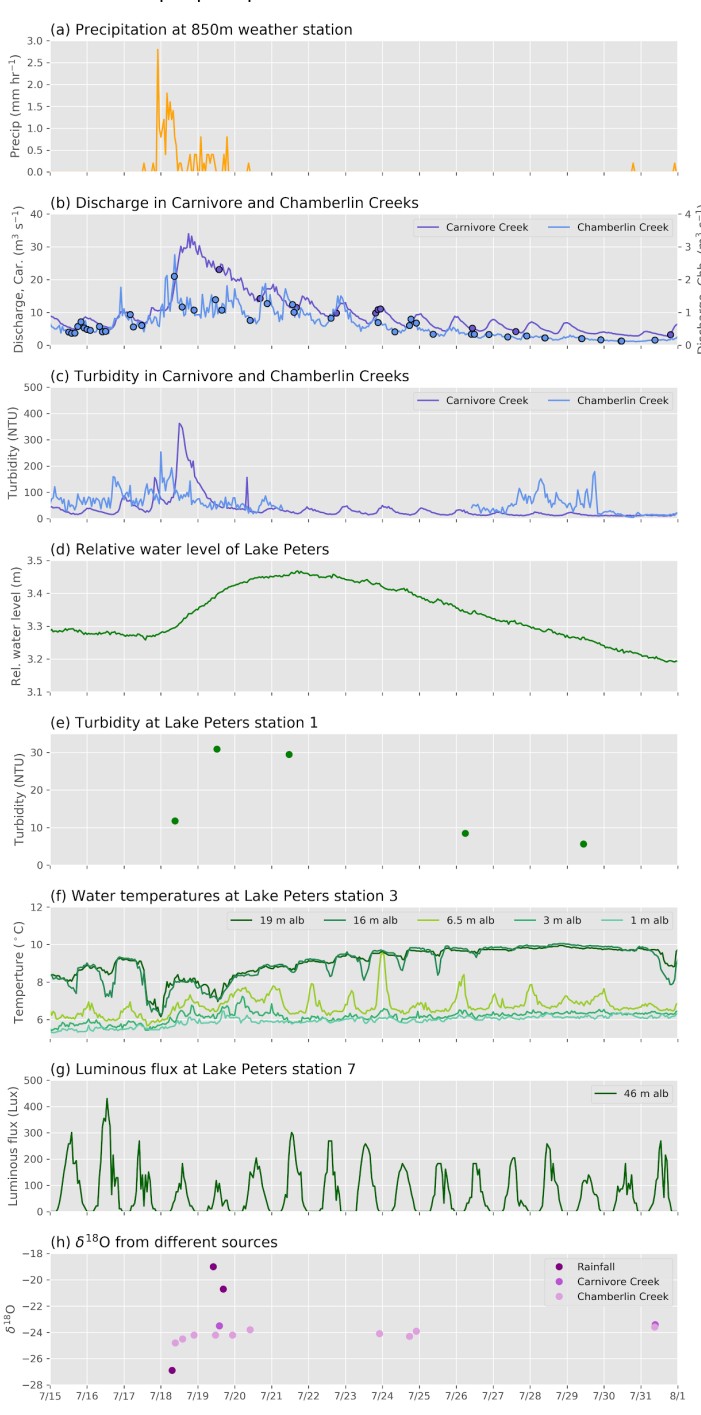



Figure 10. An example of the observatory's data, tracing the effects of a precipitation event in
July 2015. (a) Rainfall at the 850 m a.s.l. station; (b) Carnivore and Chamberlin Creek discharge
and (c) turbidity; (d) Lake Peters relative water level, (e) maximum turbidity at Lake Peters
station 1, (f) water temperatures at different heights above lake bottom (a.l.b.) at Lake Peters
station 3, and (g) luminous flux at Lake Peters station 7; and (h) $\delta^{18}O$ at different sites. To
complement panel (e), a more detailed sample of turbidity data are shown in Fig. 11.

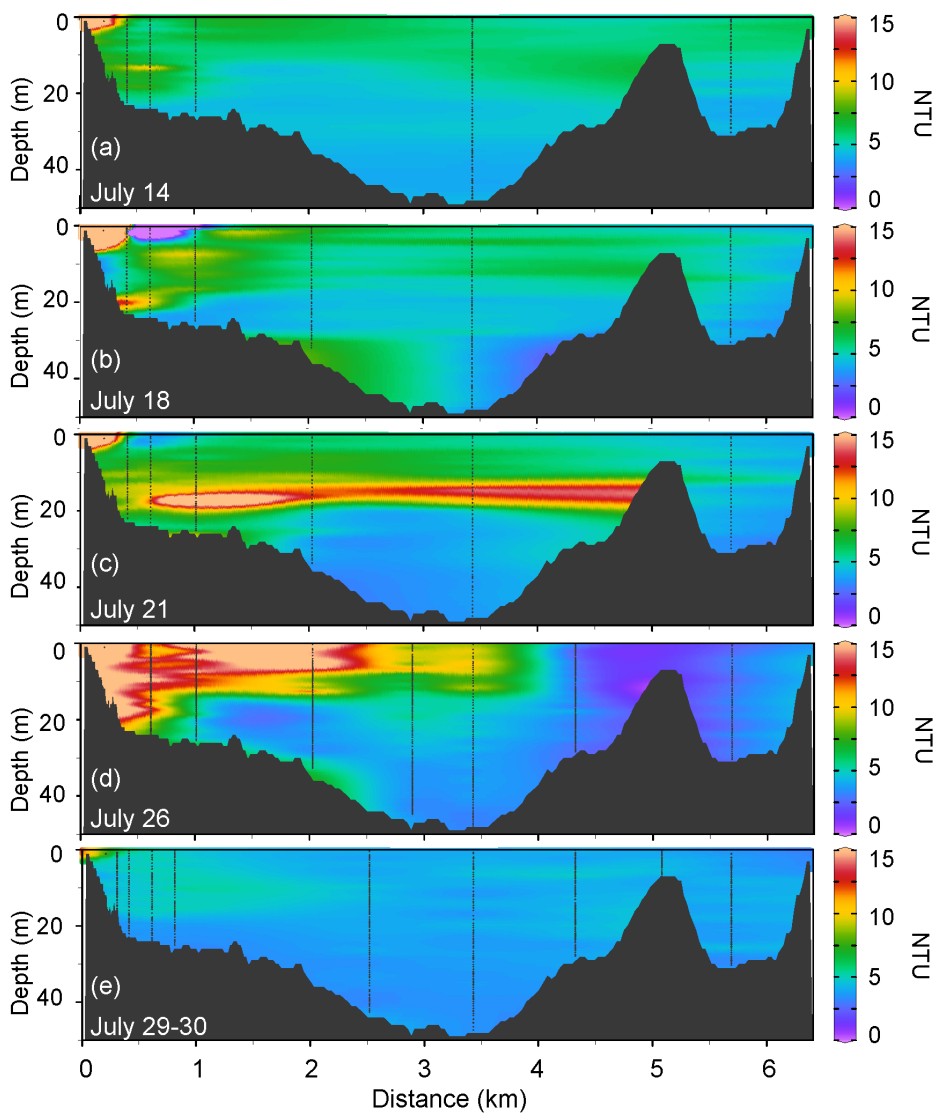


Figure 11. Turbidity in Lake Peters over the course of a precipitation event in July 2015; turbidity
data are shown for (a) July 14, (b) July 18, (c) July 21, (d) July 26, and (e) composite data from
July 29 and July 30. Black shading indicates lake bathymetry, and black dashed lines indicate
locations of TROLL casts. The x-axis indicates north-south distance across the lake from the
outflow of Carnivore Creek. Figure created using Ocean Data View (ODV) software.



# 413 5. Data availability

All DOI-referenced datasets described in this manuscript are archived at the National Science
Foundation Arctic Data Center at the following overview webpage for the project:
https://arcticdata.io/catalog/view/urn:uuid:517b8679-20db-4c89-a29c-6410cbd08afe (Kaufman
et al., 2019e). Datasets are available for download (Table 1) as csv files (or tiffs for
photogrammetry data) with accompanying metadata including an abstract, key words, methods,
spatial and temporal coverage, and personnel information pertinent to each dataset.



| Dataset | Measured variables | Coordinates | Start date | End date | DOI |
|---|---|---|---|---|---|
| 850 m a.s.l. meteorological station | barometric pressure (mbar), temperature (°C), relative humidity (%), solar radiation (W/m²), wind speed (m/s), gust speed (m/s), wind direction (ø), dew point (°C), ground temp at 2cm and 30cm (°C), precipitation (mm) | 69.3046, -145.0342 | 15-05-13 21:00 | 18-08-22 15:00 | https://doi.org/10.18739/A2V11VK4J |
| 1425 m a.s.l. meteorological station | temperature (°C), relative humidity (%), ground temp at 2 cm and 30 cm (°C), precipitation (mm) | 69.29094, -144.96983 | 15-05-24 16:00 | 18-05-26 14:00 | https://doi.org/10.18739/A2V11VK4J |
| 1750 m a.s.l. meteorological station | temperature (°C), relative humidity (%), ground temp at 2 cm and 3 0cm (°C), precipitation (mm) | 69.29043, -144.95265 | 15-06-08 00:00 | 17-08-12 11:00 | https://doi.org/10.18739/A2V11VK4J |
| Lower ablation stake | exposed stake height (cm) | 69.29312, -144.93814 | 16-04-27 13:00 | 17-08-10 13:00 | https://doi.org/10.18739/A2GQ6R21H |
| Upper ablation stake | exposed stake height (cm) | 69.29031, -144.93134 | 16-04-27 17:00 | 17-01-01 00:00 | https://doi.org/10.18739/A2GQ6R21H |
| Lower ablation stake meteorological station | temperature (°C) | 69.29312, -144.93814 | 16-04-15 5:00 | 17-08-10 23:00 | https://doi.org/10.18739/A2BZ6178M |
| Upper ablation stake meteorological station | temperature (°C), relative humidity (%) | 69.29031, -144.93134 | 16-04-15 5:00 | 17-08-10 12:00 | https://doi.org/10.18739/A2BZ6178M |
| Carnivore Creek manual data | discharge (m³/s), suspended sediment concentration (mg/L) | 69.28028, -145.03051 | 15-05-16 17:02 | 17-08-14 16:00 | https://doi.org/10.18739/A27659F58 |
| Chamberlin Creek manual data | discharge (m³/s), suspended sediment concentration (mg/L) | 2015/2017: 69.2925, -145.0261  2016: 69.2910, -145.0203 | 15-05-16 17:44 | 17-08-14 16:50 | https://doi.org/10.18739/A2MG7FV6Q |
| Carnivore Creek continuous data | conductivity (µm), temperature (°C), turbidity (NTU), pressure (m), discharge (m³/s) | 69.28028, -145.03051 | 15-05-20 16:00 | 18-08-22 13:00 | https://doi.org/10.18739/A27659F58 |
| Chamberlin Creek continuous data | conductivity (µm), temperature (°C), turbidity (NTU), pressure (m), discharge (m³/s) | 2015/2017: 69.2925, -145.0261  2016: 69.2910, -145.0203 | 15-05-22 16:00 | 17-09-23 23:00 | https://doi.org/10.18739/A2MG7FV6Q |
| Lake level | absolute (uncorrected) pressure (kPa), relative water level (m) | 69.30993, -145.04644 | 15-05-20 16:00 | 17-08-06 10:00 | https://doi.org/10.18739/A2KH0DZ5J |
| Station 3 mooring | temperature (°C) at 1, 3, 6.5, 16, and 19 meters above lake bottom | 69.29336, -145.03813 | 15-05-28 12:00 | 15-08-01 8:00 | https://doi.org/10.18739/A2FQ9Q52R |
| Station 7 mooring | temperature (°C) at 1, 3, 28, 44, and 46 meters above lake bottom; light intensity (Lux) at 28, 44, and 46 meters above lake bottom | 69.30993, -145.04644 | 15-05-15 0:00 | 16-08-05 0:00 | https://doi.org/10.18739/A2FQ9Q52R |
| Station 8 mooring | temperature (°C) at 1, 6, 11, 31, 41, and 48 meters above lake bottom; light intensity (Lux) at 48 meters above lake bottom | 69.31496, -145.04766 | 15-08-09 12:00 | 17-08-06 9:00 | https://doi.org/10.18739/A2FQ9Q52R |
| Station 11 mooring | temperature (°C) at 0.5, 3, 15, and 25 meters above lake bottom; light intensity (Lux) at 15 and 25 meters above lake bottom | 69.33472, -145.04348 | 15-05-24 0:00 | 15-08-06 12:00 | https://doi.org/10.18739/A2FQ9Q52R |
| CTD casts | temperature (°C), pressure-inferred depth (m), turbidity (NTU), conductivity (µm) | Various | 15-05-21 15:34 | 15-08-12 11:24: | https://doi.org/10.18739/A23J3912W |
| Sediment traps | mass (g), daily flux (mg/cm²/day), annual flux (g/cm²/year), organic matter content (%), grain size (µm) | Various | 15-05-25 | 17-08-06 | https://doi.org/10.18739/A2Q814S0S |
| Bathymetry | easting (UTM Zone 6N), northing (UTM Zone 6N), water depth (m) | Various | 15-08 | 15-08 | https://doi.org/10.18739/A2R785P0G |
| Photogrammetry | high resolution DEM | Various | 15-04-23 | 17-09-03 | https://doi.org/10.18739/A28W3824W |
| Stable isotopes | δ¹⁸O (‰VSMOW), δD (‰VSMOW) d-excess (‰VSMOW) | Various | 15-05-14 11:30 | 18-05-25 0:00 | https://doi.org/10.18739/A2ZS2KC8C |
| Major cation and anion geochemistry | sulfate (ppm), calcium (ppm), magnesium (ppm), potassium (ppm), sodium (ppm), nitrate (ppm), chloride (ppm) | Various | 16-08-12 18:00 | 16-08-18 20:00 | https://doi.org/10.18739/A2ZS2KC8C |


Table 1. Observational datasets generated by the Lake Peters observatory instrumentation and
archived at the NSF Arctic Data Center.

# 6. Final Remarks


The meteorological, glaciological, fluvial, and lacustrine datasets from Lake Peters and its
catchment presented in this manuscript offer a unique and valuable opportunity to study an
Arctic glacier-fed lake system. The observational data have contributed to our understanding of



individual events as well as intra- and inter-annual variability within this catchment. By making
these data available, we hope to foster a more detailed understanding of the processes leading
to sedimentation in Arctic lake catchments. Furthermore, these data may provide important
context for interpretation of lake sediment records from this study region and elsewhere, as well
as for hydrological and sedimentological modelling studies.

**Supplement**
No supplements.

**Author contributions**
N.P.M., D.S.K, E.S., D.F., J.G., M.G.L., and M.N. developed the observational design. E.B.,
L.L.T., N.P.M., D.S.K., E.S., D.F., J.G., M.G.L., M.N., S.H.A., C.W.B., R.A.E., and C.C.R. led
installation, development and/or curation of components of the observations. E.B., L.L.T.,
N.P.M., D.S.K., E.S., J.G., M.P.E., C.W., and A.J.W. wrote the original manuscript. E.B., L.L.T.,
N.P.M., E.S., D.F., S.H.A., R.A.E., M.P.E., C.W., and A.J.W. produced figures for the
manuscript. E.B., D.F., and D.S.K. curated the datasets at the NSF Arctic Data Center. All
authors provided minor edits to the text of the final manuscript.

**Competing interests**
We declare no competing interests.

**Acknowledgements**
Datasets from the Lake Peters watershed were collected with the help of numerous field
assistants, including Zak Armacost, Mindy Bell, Ashley Brown, Anne Gädeke, Jeff Gutierrez,
Rebecca Harris, Stacy Kish, Lisa Koeneman, Anna Liljedahl, Maryann Ramos, Doug Steen, and
Ethan Yackulic. Sedimentary grain size data were processed with assistance from Daniel





Cameron and Katherine Whitacre. Water isotope analyses were performed with assistance from
Jamie Brown at NAU's Colorado Plateau Stable Isotope Facility, and major cation and anion
analyses were performed by Tom Douglas of the U.S. Army Cold Regions Research and
Engineering Laboratory. We thank the U.S. Fish and Wildlife Service - Arctic National Wildlife
Refuge for use of the G. William Holmes research station and for permitting our research; Polar
Field Services, Inc./CH2MHill for support and outfitting while in the field; Dirk Nickisch and
Danielle Tirrell of Coyote Air for safely flying our field teams to and from Lake Peters;
LacCore/CSDCO for assistance with processing and archiving of sedimentary sequences from
Lake Peters; and PolarTREC for a productive partnership that contributed greatly to the broader
impacts of this project. This project was funded by NSF-1418000.


















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
