# Peer review of "An Arctic watershed observatory at Lake Peters, Alaska: weatherglacier-river-lake system data for 2015-2018"

_Earth System Science Data, 2019_

## Referee Comment (RC1) · Anonymous Referee #1 · 22 Jul 2019

Authors are commended for submitting a very polished manuscript. Before it can be accepted for publication in as-is form, they are advised to address the following minor concerns.

— In Fig. 4, the three lines (in grey, black and red) should be explained. — In section 3.3, it is described that TROLL was installed in a stage cage, which moved. Readers and potential users of the data will benefit if more discussion is added regarding how the use of the data can appropriately account for this movement. It is not clear from the provided database if the locations of the cage in different years are identified. — Presentation of Section 4 is much appreciated, especially as it highlights the tracing of

an event through the Lake Peters system. To this reviewer, the publication of data is supposed to spur/support other modeling and/or diagnostic research in the watershed. Hence, the authors are encouraged to add a (sub)section on the sufficiency of data set for hydrologic/hydraulic/water quality modeling and/or diagnostics of process interactions in the watershed. While this reviewer fully recognizes the challenges associated with collection of data in arctic watersheds, given the range of other data sets that are usually needed for modeling, it will be good for readers to recognize beforehand if the data set is "complete" to perform modeling/diagnostics. If the data set is lacking in this regard, please acknowledge the limitations and suggest ways for overcoming them. Furthermore, a discussion should be added about the possible science questions that can be answered using the presented data set, which will encourage data-use beyond the data collectors.

---

## Referee Comment (RC2) · Anonymous Referee #2 · 29 Jul 2019

The data manuscript titled "An Arctic Watershed observatory at Lake Peters, AK: weather-glacier-river-lake system data for 2015-2018" provides an interesting dataset for the four year monitoring period. Three meteorological stations provide information at three different elevations, temperature sensors observe lake temperature profiles, lake water level and sediment inputs were also documented. Additionally, stage data were collected for the two main creeks.

The manuscript is well-written and easy to follow. It was mostly clear to me what was done for sensor installation and how the data were processed, with some minor gaps further detailed below. My primary concern with the data is the hydrological data. It is

my understanding that the location that stage data were collected changed from year to year and all data were attempted to be corrected to an initial rating curve developed in 2015. This can not be done. The authors state the assumption of a consistent stage-discharge relation, but this assumption can only be made if stage is taken at the exact same location AND the cross section of the channel does not change at that location from year to year (as well as immediately up and down stream). Based on the data presented, these assumptions can not be made and thus the only discharge data available and reliable are for 2015. When downloading these hydrological data I was also surprised to only find the discharge data, it would be helpful for data users to have access to stage data and specific location information. It is my recommendation that only stage data are made available for all years other than 2015, and for 2015 both stage and discharge data with an explicit statement in the metadata about the moving of pressure transducer and differing cross sections.

Additionally, the application of the dataset is certainly interesting. However, I would like to see a bit more details and analysis of all these data. Perhaps an example of a simple model, as mentioned, and comparison to the flood event. Or perhaps a detailed analysis and inventory of the dataset.

I am recommending major changes based on my concerns with the hydrological data and desire to see more analysis. Minor comments are listed below and referred to by approximate line number.

33: I suggest removing webpage link and citation from the abstract.

55: What implications do the "non-permanent" installations have on the dataset

Fig. 4: The gray dots and text is quite difficult to see and read, I suggest a different color.

186: How often were these measurements taken? How many points in total did you use for the rating curve?
202: This assumption is really based on stage being measured at the same location and channel X-section not changing which doesn't appear to be true here.

Fig. 7: the panel identifiers a-j are difficult to see, maybe make them bigger and in a top corner?

---

## Author Comment (AC2) · 17 Sep 2019

We thank anonymous Referee 2 for offering constructive feedback on our manuscript. Below we have indicated the actions we plan to take to address each of the items noted by this reviewer. Referee text (R) and author responses (A) are indicated.

R: The data manuscript titled "An Arctic Watershed observatory at Lake Peters, AK: weather-glacier-river-lake system data for 2015-2018" provides an interesting dataset for the four year monitoring period. Three meteorological stations provide information at three different elevations, temperature sensors observe lake temperature profiles, lake water level and sediment inputs were also documented. Additionally, stage data

were collected for the two main creeks. The manuscript is well-written and easy to follow. It was mostly clear to me what was done for sensor installation and how the data were processed, with some minor gaps further detailed below. My primary concern with the data is the hydrological data. It is my understanding that the location that stage data were collected changed from year to year and all data were attempted to be corrected to an initial rating curve developed in 2015. This can not be done. The authors state the assumption of a consistent stage-discharge relation, but this assumption can only be made if stage is taken at the exact same location AND the cross section of the channel does not change at that location from year to year (as well as immediately up and down stream). Based on the data presented, these assumptions can not be made and thus the only discharge data available and reliable are for 2015.

A: We appreciate this reviewer's concerns with the hydrological data for 2016, 2017, and 2018. For 2015, we applied standard stage-discharge rating curve methods, with extensive hydrologic field data spanning nearly the full open-water period and discharge range, allowing robust calculation of discharge uncertainty. We acknowledge the complications experienced in subsequent years make the hydrologic data less reliable than what is typical for published discharged records. Most substantial was the temporary gauge relocation in 2016 for Chamberlin Creek, but also potential shifting rating curve relations and some instrumentation failures.

Despite the acknowledged limitations in the hydrologic data for 2016 onward, we believe that our discharge record is of value as being indicative or approximate of hydrologic conditions in this region where little such hydrologic data exist. Significant effort was made to collect reasonably comparable data among years and we observe relatively strong relations among our discharge measures for all years with stage, cross-sectional geometry, and photographed water levels (Fig. 7 e-j; 2017 CAR stage-Q $R^2$=.996 n=4; 2017 CHB stage-Q $R^2$=.881 n=13).

To avoid misrepresenting data quality and identify methodological changes past 2015, we propose to report and illustrate these data to be approximate in the text, plots, and

datasets. For the metadata in the Arctic Data Center, we will code all discharge observations to flag where gauging locations were relocated, where new rating curves were developed and used, where rating curve relations were extrapolated from a preceding year, and where photograph-based interpolations were used, and these will be added to the existing metadata for these datasets.

R: When downloading these hydrological data I was also surprised to only find the discharge data, it would be helpful for data users to have access to stage data and specific location information. It is my recommendation that only stage data are made available for all years other than 2015, and for 2015 both stage and discharge data with an explicit statement in the metadata about the moving of pressure transducer and differing cross sections.

A: We will add stage data for all years to the datasets at the Arctic Data Center, as recommended. See comments above on the inclusion of discharge data for 2016-2018 as approximate data.

R: Additionally, the application of the dataset is certainly interesting. However, I would like to see a bit more details and analysis of all these data. Perhaps an example of a simple model, as mentioned, and comparison to the flood event. Or perhaps a detailed analysis and inventory of the dataset.

A: We agree that these datasets have great potential for a variety of modeling applications in Earth system science. However, because this is a data description paper, we have refrained from including any analyses, as these would be beyond the scope of this manuscript. We included section 4 as an example of the utility of the data, but intentionally refrained from interpreting this example, an approach we believe is appropriate for a data description paper. We will include an additional subsection in section 4 that suggests limitations and potential uses of these data for modeling applications. For an inventory of the data, please refer to Figure 1 and Table 1.

R: I am recommending major changes based on my concerns with the hydrological

data and desire to see more analysis. Minor comments are listed below and referred to by approximate line number. 33: I suggest removing webpage link and citation from the abstract.

A: The webpage link and citation are included in the abstract per the guidelines of this journal and a direct request from the editors to do so.

R: 55: What implications do the "non-permanent" installations have on the dataset.

A: We agree that the implications of this descriptor might be unclear. A few words clarifying the meaning of "non-permanent" installations will be added.

R: Fig. 4: The gray dots and text is quite difficult to see and read, I suggest a different color.

A: We agree that this color scheme could be improved for clarity. The color scheme will be changed from black and grey to black, red, and blue.

R: 186: How often were these measurements taken? How many points in total did you use for the rating curve?

A: We agree that the frequency and number of datapoints used to create the rating curve should be explicitly stated. Clarifying text will be added.

R: 202: This assumption is really based on stage being measured at the same location and channel X-section not changing which doesn't appear to be true here.

A: See comments above on further justification added to address these concerns regarding the hydrological data, particularly the coding/flagging of gauge relocations.

R: Fig. 7: the panel identifiers a-j are difficult to see, maybe make them bigger and in a top corner?

A: We agree that the panel identifiers should be altered, and will do so as described by the reviewer.

For a formatted version of this comment, please see the supplementary PDF.

Please also note the supplement to this comment:
https://www.earth-syst-sci-data-discuss.net/essd-2019-60/essd-2019-60-AC2-supplement.pdf

---

## Author Response (AR1)

We thank anonymous Referee 1 for offering constructive feedback on our manuscript. Below we have indicated the actions we plan to take to address each of the items noted by this reviewer. Referee text is italicized and author responses are bolded.

Authors are commended for submitting a very polished manuscript. Before it can be accepted for publication in as-is form, they are advised to address the following minor concerns. In Fig. 4, the three lines (in grey, black and red) should be explained.

We agree that this figure caption should be more explicit. A more detailed caption will be included to explain the purpose of the three lines in Figure 4. The red line is the 1:1 correlation line for reference, while the black and grey lines are the correlations between precipitation recorded at the 850 m meteorological station and precipitation recorded at the 1425 m station (black) and 1750 m station (grey).

In section 3.3, it is described that TROLL was installed in a stage cage, which moved. Readers and potential users of the data will benefit if more discussion is added regarding how the use of the data can appropriately account for this movement. It is not clear from the provided database if the locations of the cage in different years are identified.

We concur that the statement that the cage shifted several times in Chamberlin Creek is unclear, without providing context regarding how this affected our dataset. We will add clarifying information to Section 3.3 of the manuscript. We also provide explanation here, for your information. In Carnivore Creek, there was no significant shifting of the TROLL and Hobo sensors attached to the cage, except that instruments failed during the August 2015 flood, at which time data were downloaded. While the TROLL water pressure data were used for Carnivore Creek, the TROLL shifted in Chamberlin Creek (as the reviewer noted), and therefore the Hobo data provided a more complete record at Chamberlin Creek. The Hobo instrument in Chamberlin Creek did shift on one occasion on 06/04/2015, and was reestablished in a slightly different position on 06/05/2015. We corrected for the small shift in instrument position by comparing water pressure from the Hobo and TROLL and adjusting the early season Hobo water pressure to match data from 06/05/2015 through to the end of the season. Data from the TROLL instrument, which shifted several times in 2015, were not used, other than the subset of data that was used to correct the Hobo water pressure. Further detail will be added to Section 3.3 of the manuscript to clarify the quality of the hydrological data given that the Chamberlin Creek hydrological station (TROLL and Hobo) was secured in a different location in 2016. The coordinates of the hydrological stations for each year are given in the Arctic Data Center link for this dataset, and this will be clarified in the manuscript.

Presentation of Section 4 is much appreciated, especially as it highlights the tracing of an event through the Lake Peters system. To this reviewer, the publication of data is supposed to spur/support other modeling and/or diagnostic research in the watershed. Hence, the authors are encouraged to add a (sub)section on the sufficiency of data set for hydrologic/hydraulic/water quality modeling and/or diagnostics of process interactions in the watershed. While this reviewer fully recognizes the challenges associated with collection of data in arctic watersheds, given the range of other data sets that are usually needed for modeling, it will be good for readers to recognize beforehand if the data set is "complete" to perform modeling/diagnostics. If the data set is lacking in this regard, please acknowledge the limitations and suggest ways for overcoming them. Furthermore, a discussion should be added about the possible science questions that can be answered using the presented data set, which will encourage data-use beyond the data collectors.

We agree that one of the primary goals of these datasets (although not the only objective) is to support modelling efforts in the watershed and region. As suggested, we will add a subsection to section 4 that describes how the data can be used for hydrologic, sediment transport and sediment deposition modeling, including the shortcomings of the datasets and suggested methods for overcoming these limitations, as well as possible science questions that can be answered using the datasets. We thank anonymous Referee 2 for offering constructive feedback on our manuscript. Below we have indicated the actions we plan to take to address each of the items noted by this reviewer. Referee text is italicized and author responses are bolded.

The data manuscript titled "An Arctic Watershed observatory at Lake Peters, AK: weatherglacier-river-lake system data for 2015-2018" provides an interesting dataset for the four year monitoring period. Three meteorological stations provide information at three different elevations, temperature sensors observe lake temperature profiles, lake water level and sediment inputs were also documented. Additionally, stage data were collected for the two main creeks. The manuscript is well-written and easy to follow. It was mostly clear to me what was done for sensor installation and how the data were processed, with some minor gaps further detailed below. My primary concern with the data is the hydrological data. It is my understanding that the location that stage data were collected changed from year to year and all data were attempted to be corrected to an initial rating curve developed in 2015. This can not be done. The authors state the assumption of a consistent stage-discharge relation, but this assumption can only be made if stage is taken at the exact same location AND the cross section of the channel does not change at that location from year to year (as well as immediately up and down stream). Based on the data presented, these assumptions can not be made and thus the only discharge data available and reliable are for 2015.

We appreciate this reviewer's concerns with the hydrological data for 2016, 2017, and 2018. For 2015, we applied standard stage-discharge rating curve methods, with extensive hydrologic field data spanning nearly the full open-water period and discharge range, allowing robust calculation of discharge uncertainty. We acknowledge the complications experienced in subsequent years make the hydrologic data less reliable than what is typical for published discharged records. Most substantial was the temporary gauge relocation in 2016 for Chamberlin Creek, but also potential shifting rating curve relations and some instrumentation failures.

Despite the acknowledged limitations in the hydrologic data for 2016 onward, we believe that our discharge record is of value as being indicative or approximate of hydrologic conditions in this region where little such hydrologic data exist. Significant effort was made to collect reasonably comparable data among years and we observe relatively strong relations among our discharge measures for all years with stage, cross-sectional geometry, and photographed water levels (Fig. 7 e-j; 2017 CAR stage-Q R2=.996 n=4; 2017 CHB stage-Q R2=.881 n=13).

To avoid misrepresenting data quality and identify methodological changes past 2015, we propose to report and illustrate these data to be approximate in the text, plots, and datasets. For the metadata in the Arctic Data Center, we will code all discharge observations to flag where gauging locations were relocated, where new rating curves were developed and used, where rating curve relations were extrapolated from a preceding year, and where photograph-based interpolations were used, and these will be added to the existing metadata for these datasets.

When downloading these hydrological data I was also surprised to only find the discharge data, it would be helpful for data users to have access to stage data and specific location information. It is my recommendation that only stage data are made available for all years other than 2015, and for 2015 both stage and discharge data with an explicit statement in the metadata about the moving of pressure transducer and differing cross sections. We will add stage data for all years to the datasets at the Arctic Data Center, as recommended. See comments above on the inclusion of discharge data for 2016-2018 as approximate data.

Additionally, the application of the dataset is certainly interesting. However, I would like to see a bit more details and analysis of all these data. Perhaps an example of a simple model, as mentioned, and comparison to the flood event. Or perhaps a detailed analysis and inventory of the dataset.

We agree that these datasets have great potential for a variety of modeling applications in Earth system science. However, because this is a data description paper, we have refrained from including any analyses, as these would be beyond the scope of this manuscript. We included section 4 as an example of the utility of the data, but intentionally refrained from interpreting this example, an approach we believe is appropriate for a data description paper. We will include an additional subsection in section 4 that suggests limitations and potential uses of these data for modeling applications. For an inventory of the data, please refer to Figure 1 and Table 1.

*I am recommending major changes based on my concerns with the hydrological data and desire to see more analysis. Minor comments are listed below and referred to by approximate line number. 33: I suggest removing webpage link and citation from the abstract.*

The webpage link and citation are included in the abstract per the guidelines of this journal and a direct request from the editors to do so.

55: What implications do the "non-permanent" installations have on the dataset.

We agree that the implications of this descriptor might be unclear. A few words clarifying the meaning of "non-permanent" installations will be added.

Fig. 4: The gray dots and text is quite difficult to see and read, I suggest a different color.

We agree that this color scheme could be improved for clarity. The color scheme will be changed from black and grey to black, red, and blue.

186: How often were these measurements taken? How many points in total did you use for the rating curve?

We agree that the frequency and number of datapoints used to create the rating curve should be explicitly stated. Clarifying text will be added.

202: This assumption is really based on stage being measured at the same location and channel X-section not changing which doesn't appear to be true here.

See comments above on further justification added to address these concerns regarding the hydrological data, particularly the coding/flagging of gauge relocations.

Fig. 7: the panel identifiers a-j are difficult to see, maybe make them bigger and in a top corner?

We agree that the panel identifiers should be altered, and will do so as described by the reviewer.

**An Arctic watershed observatory at Lake Peters, Alaska: weather-glacier-river-lake system data for 2015-2018**

**Authors**

Ellie Broadman1, Lorna L. Thurston2, Erik Schiefer1, Nicholas P. McKay1, Darrell S. Kaufman4, Erik Schiefer4, David Fortin3, Jason Geck4, Michael G. Loso5, Matt Nolan6, Stéphanie H. Arcusa1, Christopher W. Benson1, Rebecca A. Ellerbroek1, Michael P. Erb1, Cody C. Routson1, Charlotte Wiman1, A. Jade Wong1, Darrell S. Kaufman1

[revised manuscript text omitted]